# Starve a cold or feed a fever? Identifying cellular metabolic changes following infection and exposure to SARS-CoV-2

Emma K. Loveday[ORCID][1,2,3]*, Hope Welhaven[4], Ayten Ebru Erdogan[ORCID][2], Kyle S. Hain[3], Luke F. Domanico[3], Connie B. Chang[ORCID][1,2,5], Ronald K. June[ORCID][6], Matthew P. Taylor[ORCID][3]*

1 Center for Biofilm Engineering, Montana State University, Bozeman, Montana, United States of America, 2 Department of Chemical and Biological Engineering, Montana State University, Bozeman, Montana, United States of America, 3 Department of Microbiology and Cell Biology, Montana State University, Bozeman, Montana, United States of America, 4 Chemistry and Biochemistry, Montana State University, Bozeman, Montana, United States of America, 5 Department of Physiology and Biomedical Engineering, Mayo Clinic, Rochester, Minnesota, United States of America, 6 Department of Mechanical & Industrial Engineering, Montana State University, Bozeman, Montana, United States of America

* emma.loveday@montana.edu(EKL); mptaylor@montana.edu(MPT)

## Abstract

Viral infections induce major shifts in cellular metabolism elicited by active viral replication and antiviral responses. For the virus, harnessing cellular metabolism and evading changes that limit replication are essential for productive viral replication. In contrast, the cellular response to infection disrupts metabolic pathways to prevent viral replication and promote an antiviral state in the host cell and neighboring bystander cells. This competition between the virus and cell results in measurable shifts in cellular metabolism that differ depending on the virus, cell type, and extracellular environment. The resulting metabolic shifts can be observed and analyzed using global metabolic profiling techniques to identify pathways that are critical for either viral replication or cellular defense. SARS-CoV-2 is a respiratory virus that can exhibit broad tissue tropism and diverse, yet inconsistent, symptomatology. While the factors that determine the presentation and severity of SARS-CoV-2 infection remain unclear, metabolic syndromes are associated with more severe manifestations of SARS-CoV-2 disease. Despite these observations a critical knowledge gap remains between cellular metabolic responses and SARS-CoV-2 infection. Using a well-established untargeted metabolomics analysis workflow, we compared SARS-CoV-2 infection of human lung carcinoma cells. We identified significant changes in metabolic pathways that correlate with either productive or non-productive viral infection. This information is critical for characterizing the factors that contribute to SARS-CoV-2 replication that could be targeted for therapeutic interventions to limit viral disease.

## Introduction

As obligate intracellular parasites, viruses co-opt host cellular materials, machinery, and metabolism to facilitate viral replication [1,2]. Metabolic changes in response to virus infection result in extensive alterations to cellular physiology and often mirror changes seen in

**Data availability statement:** All relevant data are within the paper and its Supporting information files.

**Funding:** RKJ NIH/NIAMS R01AR073964 and R01AR081489 National Institutes of Health, National Institute of Arthritis, Musculoskeletal, and Skin Diseases https://www.niams. nih.gov/ No role RKJ NSF CMMI 1554708 National Science Foundation, Division of Civil, Mechanical and Manufacturing Innovation https://www.nsf.gov/div/index.jsp?div=CMMI no role CBC NIH/NIAID 1R56AI156137-01 National Institutes of Health, National Institute of Allergy and Infectious Disease https://www. niaid.nih.gov/ No role MPT Montana State University VPREDGE COVID Research Award Internal Award https://www.montana.edu/ research/ No role

**Competing interests:** The authors have declared that no competing interests exist.

cancer cells [2–7]. Both RNA and DNA viruses reprogram different aspects of host metabolism including increased glycolysis, elevated pentose phosphate activity, amino acid generation and lipid synthesis [1]. Viral hijacking of host metabolism and subversion of metabolic defenses can lead to increased viral replication and host damage, resulting in long-term health consequences, such as those seen in severe cases of COVID-19, following infection with the novel coronavirus, SARS-CoV-2. This is supported by analysis of SARS-CoV-2 positive patient serum that has shown acute and long-term changes in metabolites and further metabolic disorder [8–12].

To better understand what metabolic changes occur during SARS-CoV-2 infection and how this may relate to severe disease outcomes, we implemented global metabolomic profiling to analyze thousands of metabolites using LC-MS to detect disease-associated changes to the cellular environment [13,14]. Metabolites serve as intermediates for cellular physiology and include hormones, oligonucleotides, peptides, and other molecular products of cellular biochemical reactions that represent the current physiological state of a cell [15]. Global metabolomic profiling can therefore provide an unbiased view of metabolic shifts induced during and in response to viral infection [16,17].

To elucidate changes to cellular metabolism associated with SARS-CoV-2 viral replication and those changes associated with virus exposure we infected and profiled A549 cells, a human lung cell line. A549 cells are frequently used to evaluate viral infection for many respiratory viruses but are not intrinsically susceptible to SARS-CoV-2 infection, as they lack endogenous expression of the viral receptor, ACE2 [18–20]. However, expression of human ACE2 protein on A549 (ACE2-A549) cells renders them fully susceptible to SARS-CoV-2 [21]. By comparing A549 and ACE2-A549 cells inoculated with SARS-CoV-2 we can identify and separate metabolic shifts induced by active viral replication from those induced by the host cells response to virus exposure.

Here, we describe distinct metabolic changes in to both ACE2-A549 and A549 cells triggered by SARS-CoV-2 exposure. Amino acid metabolism, glutathione, and urea cycle metabolic pathways were significantly altered in cells that support productive SARS-CoV-2 infection (ACE2-A549 cells). In contrast A549 cells that are not susceptible to infection but were exposed to a high inoculating dose of SARS-CoV-2 had significant changes in fatty acid anabolic and catabolic pathways as well as leukotriene metabolism. These results mirror the metabolic shifts found in serum from patients suffering from severe COVID-19 [10,11,17,22–27]. Thus, our findings point to metabolite features associated with both active infection and exposure to virus. Understanding how cellular metabolism is reprogrammed following SARS-CoV-2 infection will allow identification of factors responsible for severe disease and aid in the development of antiviral therapies.

## Materials and methods

### Cells and viruses

E6-Vero, A549, ACE2-A549 cells. E6 Vero cells were obtained from ATCC (Manassas, VA) and grown in DMEM supplemented with 10% FBS, 1% pen-strep. A549 cells were the obtained from Chang Lab. ACE2-A549 cells were obtained from BEI Resources (NR-53821). A549 cells were propagated in Hams F-12 (Corning) media supplemented with 10% fetal bovine serum (HyClone) and 1X Penicillin/Streptomycin (Fisher Scientific). ACE2-A549 cells were supplemented with 100ug/mL Blasticidin (Gibco). SARS-CoV-2 strain WA01 was obtained from BEI Resources (NR-52281). Viral stocks were propagated and titered on E6 Vero cells in DMEM supplemented with 2% FBS and 1% pen-strep. Viral stocks were made by collecting media from infected cell cultures showing extensive cytopathic effect and

centrifuged 1,000 RCF for 5 minutes to remove cellular debris. The clarified viral supernatant was then used for all experimental infection. For determination of viral infectivity by plaque assay, E6 Vero cells were cultured then incubated with viral inoculum at limiting dilutions. Dilutions employed resulted in a minimum threshold of detection at 500 plaque forming units (PFU)/mL. Following inoculation, cells were over-layered with 1% methylcellulose, DMEM supplemented with 2% FBS and 1% pen-strep and incubated for 3-4 days [28,29]. Cells were then fixed and stained with 0.5% methylene blue/70% ethanol solution. Plaques were counted and the overall titer was calculated.

## Western blot detection

Both A549 or ACE2-A549 cells were cultured and collected with PBS supplemented with 0.5 mM EDTA. Cells were pelleted by centrifugation then resuspended in protein extraction buffer [10 mM Tris–HCl (pH 7.5), 10 mM NaCl, 1.5 mM MgCl2, 1% NP40] supplemented with Mini-complete, EDTA Free Protease Inhibitor Cocktail (Roche Applied Science, Indianapolis, ID). Protein concentrations determined by BCA assay, and then 20 μg were loaded and separated on a 10% polyacrylamide gel. Subsequently, proteins were transferred to PVDF membranes and incubated with a goat anti-human ACE2 mAb (R&D Systems Cat # AF933) and secondary rabbit-anti goat AF488-conjugated mAb (ThermoFisher, Cat # A27012). After detection, the blot was stripped and re-probed with mouse anti-beta Actin and a goat anti-mouse DyLight550 secondary (ThermoFisher, Cat # 84540). Membranes were imaged on a Cytiva Amersham Typhoon scanner (Bucks, UK) using the Cy2 and Cy3 channels.

## Immunofluorescence detection of infection

Prior to infection, cells were seeded at 2 x 10$^4$ per well of an eight-chamber coverslip (Labtek Cat. No. 155411, Nunc International, Rochester, NY). At indicated times post infection, cells were then fixed with 4% paraformaldehyde in PBS for 30 minutes, washed thoroughly with PBS, and blocked in 2% bovine serum albumin (BSA) prior to antibody incubations. Primary and secondary antibodies were diluted in a PBS supplemented with 0.5% saponin, 0.125% BSA as described [30], and incubated for one hour at room temperature. Primary mouse anti-nucleocapsid (Thermofisher, Cat # MA1-7403) was diluted 1:500, followed by goat anti-mouse IgG labeled with Dylight 550 ThermoFisher, Cat # 84540) at 1:500. DNA was counterstained with Hoescht 3342 at 1:5000 dilution. Actin filaments stained with phalloidin-488 (Thermofisher, Cat # A12379) at 1:500. Stained cells were imaged on a Nikon Ti-Eclipse inverted epifluorescent microscope (Nikon Instruments, Melville, NY) equipped with an iXon 896 EM-CCD (Andor Technology Ltd., Belfast, Northern Ireland) camera. Fluorescence detection used a SpectraX LED light engine (Lumencor, Beaverton, OR) with paired excitation filters, dichroic mirrors, and emission filters (Prior Scientific, Rockland, MA). Images were acquired with either Plan Fluor 20 phase contrast (Ph) air objective or CFI Plan Apochromat Lambda 60x Oil immersion objective. All imaging experiments were performed a minimum of two times.

## Metabolite extractions

ACE2-A549 and A549 cells were cultured in 6-well plates to approximately 90% confluency. Cells were then inoculated with SARS-CoV-2 for one hour, washed with PBS, then fed with fresh media. Cells were harvested at 0-, 6-, and 16-hours post-inoculum removal with 6 replicate wells harvested separately at each time point. At each collection, cells were washed with PBS, suspended with trypsin-EDTA for 5 minutes, collected and centrifuged for 5 minutes. Trypsin-EDTA was removed, and cell pellets were washed with an equi-volume of PBS before

repeated centrifugation. PBS was removed and cells were resuspended in 100% methanol. Samples were vortexed in 10 x 1 sec bursts before being placed in −80 °C freezer. Vortexing and freezing was repeated 3 times to maximize macromolecule precipitation. Subsequently, methanol extracts were subjected to centrifugation at 8,000 rcf for 10 minutes to pellet cell debris and precipitate proteins. The supernatant containing the metabolites was transferred to a separate tube and dried by vacuum concentration to remove solvents. Dried metabolites were resuspended in 100 μL mass spectrometry grade 50:50 (v/v) water: acetonitrile solution immediately prior to high performance liquid chromatography-mass spectrometry (HPLC-MS) analysis.

### Untargeted metabolomic analysis

Extracted metabolites were analyzed using HPLC-MS (Agilient 6538 Q-TOF mass spectrometer) in positive mode (resolution: ~ 20 ppm, accuracy: ~ 5 ppm, possible ionization adducts: $H^+$, $Na^+$) using a Cogent Diamond Hydride HILIC column (150 x 2.1 mm). LC-MS data, consisting of mass-to-charge (m/z) values and their peak intensities, were processed and exported using MSConvert and XCMS (S1 Table). All data was log transformed and autoscaled prior to analysis using MetaboAnalyst [31–33].

Statistical analyses performed included hierarchical cluster analysis (HCA), principal component analysis (PCA), partial least-squares discriminant analysis (PLS-DA), variable importance in projection (VIP) scores, volcano plot, fold change, and heatmap analysis. Pathway analysis was performed to map differentially expressed metabolite features to biological pathways using the Functional Analysis function in MetaboAnalyst (pathway library: KEGG, mass tolerance: 5 ppm, positive mode) [31,32]. Pathway significance was determined using FDR-corrected significance levels of 0.05.

For metabolomic data and downstream pathway analyses, there were 35 samples total (6 samples per timepoint for each cell line except for only 5 samples for the t6 timepoint in ACE2 cells). In total, 1,085 metabolite features were co-detected across all samples. To examine differences in regulation patterns across timepoints, ANOVA analysis was performed. The results of this analysis are that 152 and 372 metabolite features were differentially regulated across timepoints in ACE2-A549 cells and A549 cells alone, respectively. From here, we took these differentially-regulated features and performed pathway enrichment analyses. Thus 152 features were used for ACE2-A549 pathways and 372 features for A549 pathways. For features that are differentially regulated across timepoints of ACE2-A549 cells, 13 pathways were identified. Conversely, 8 pathways were differentially regulated across timepoints of A549 cells.

## Results

### Differing susceptibility and productivity of A549 cells lines for SARS-CoV-2

To study metabolic shifts during SARS-CoV-2 infection, we employed A549 cells, a human lung carcinoma cell line. A549 cells are not susceptible to SARS-CoV2 infection and must be modified to express human ACE2 to allow for entry and replication [18–21]. To confirm human ACE2 expression, cell extracts from A549 cells engineered to express ACE2 (ACE2-A549) and the progenitor A549 cells were subjected to western blot detection (Fig 1A).

To evaluate the capacity to support SARS-CoV-2 infection, ACE2-A549 and A549 cells were infected and analyzed to explore differences in susceptibility and relative timing of viral replication. Our analysis focused on 6 hpi, a timepoint suggested to be prior to the onset of progeny virus production, and 16 hpi, when productive viral replication should be close to its peak [34]. Infections were performed at an MOI of 10 to ensure homogenous infection and

exposure of all cells to sufficient infectious inoculum. The extent of viral infection and replication was analyzed using indirect immunofluorescent detection of the SARS-CoV-2 nucleocapsid (N) protein (Fig 1B). ACE2-A549 cells displayed extensive N protein expression at both 6 and 16 hpi. In contrast, unmodified A549 cells displayed no N protein staining, indicating a complete lack of infection and replication following SARS-CoV-2 inoculation. An important element to analyzing metabolic profiles is the relative "health" of the cell, especially at later time points in the viral lifecycle. To evaluate whether 16 hpi exhibits extensive cell deterioration, we analyzed the distribution of actin filaments in infected cells. Cells were counterstained with both SARS-CoV-2 nucleocapsid (N) protein and phalloidin to image for the presence

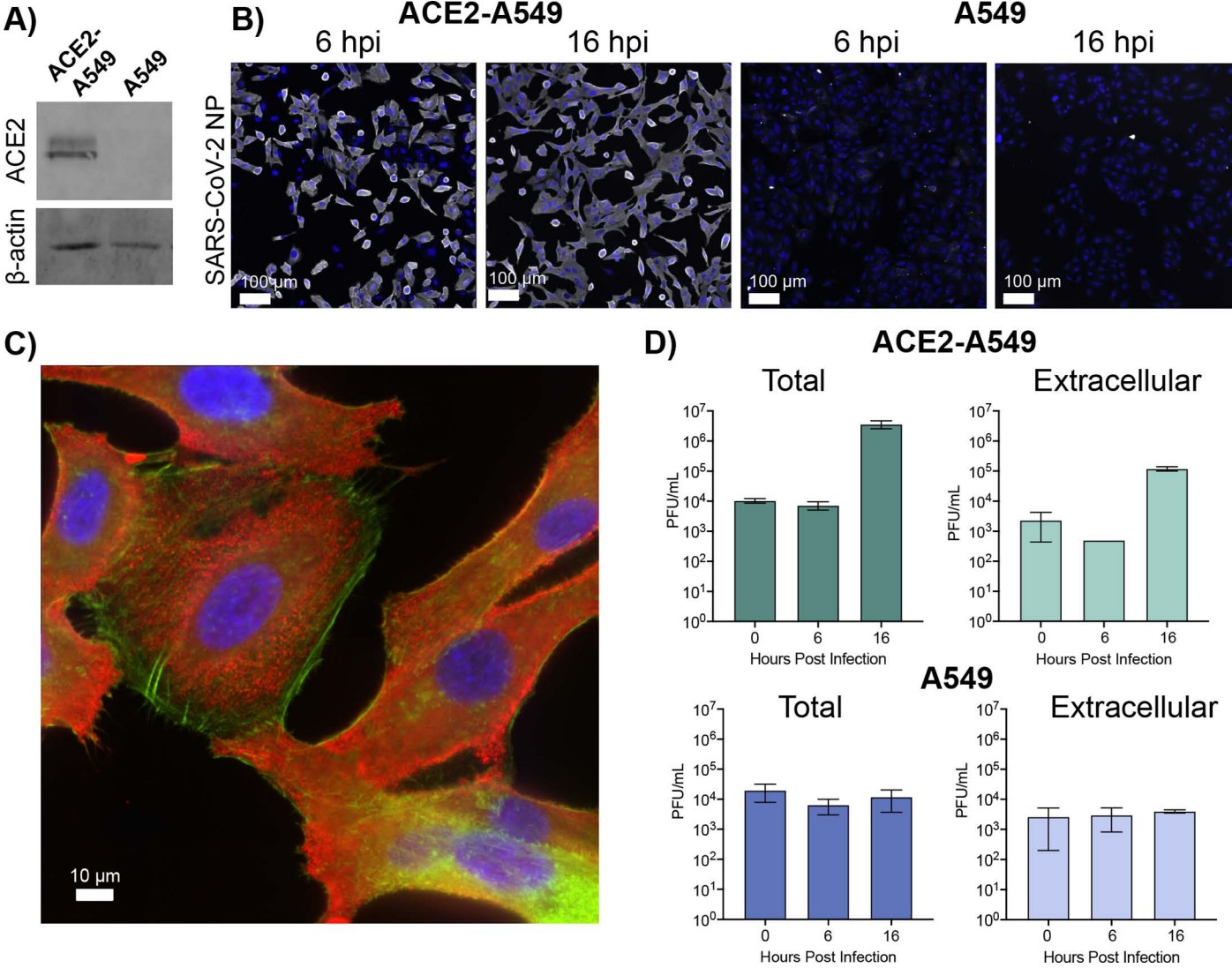

**Fig 1. Infection of different A549 cells with SARS-CoV-2.** (A) Extracts from ACE2 expressing or parental A549 cells (20 μg protein per lane) were separated by SDS-PAGE and transferred for western blotting against ACE2 and beta-actin, respectively. (B) ACE2-A549 and A549 cells were infected at MOI 10 with SARS-CoV-2/WA01. Parallel infections were fixed at 6 and 16 hpi and stained with SARS-CoV-2 anti-nucleocapsid antibody (white) with Dapi (blue, nuclei). Scale bar is 100 μm. (C) Co-staining of the SARS-CoV-2 anti nucleocapsid antibody (red) with phalloidin (green) with Dapi (blue, nuclei) in ACE2-A549 cells at 16 hpi at MOI of 10. (D) Titers from ACE2-A549 and A549 cells infected with SARS-CoV-2 at an MOI 10. Both cell extracts (Total) and supernatants (Extracellular) were collected and titered at 0, 6, and 16 hpi. All data represented as the mean ± SD of three independently infected samples.

of actin filament assemblies during infection (Fig 1C). We observed a distribution of cellular morphologies, but many cells retain actin filament assemblies and adhesions to the cell surface similar to the A549 cells that do not support productive replication (S1 Fig). The distribution of cellular morphology and actin staining suggests that the cells are not undergoing extensive cytopathic effect by 16 hpi.

To understand how the selected timepoints correlate with viral replication, we analyzed viral titers from supernatant with (total) and without (extracellular) cellular fractions, (Fig 1D). We compared the detection of plaque forming units (PFU) from ACE2-A549 and A549 cells at all three timepoints. In the ACE2-A549 cells, viral titer does not increase until 16 hours post inoculation in both the total and extracellular samples. A decrease at 6 hpi in the extracellular samples reflects the uptake of viral inoculum. The subsequent increase in titer at 16 hpi correlates with the release of virus. For A549 cells there was no increase in viral titer in either the total or extracellular samples beyond what is detected after inoculation of cells. These observations are consistent with the reported lack of susceptibility and permissiveness of A549 cells to SARS-CoV-2 [19,20]. The differences in infection, replication, and release of infectious virus supports our selection in timepoints to explore metabolic changes in cells that can support robust productive virus infection and in cells that do not allow viral entry.

## Experimental design to assess metabolic differences following SARS-CoV-2 infection

Global metabolomic profiling was performed in cells infected with or exposed to SARS-CoV-2. Our analysis focuses on three critical timepoints: 0 hours post infection (hpi), or immediately after inoculum removal, 6 hpi and 16 hpi [34]. In our experimental approach, ACE2-A549 and A549 cells were inoculated at an MOI of 10 with SARS-CoV-2 to ensure homogeneity across the population of cells critical for the subsequent extraction and detection of metabolites. The cells were collected and processed at 0, 6, and 16 hpi, to analyze temporal changes in the metabolic landscape over the course of the viral lifecycle (Fig 2A). Metabolites were extracted and

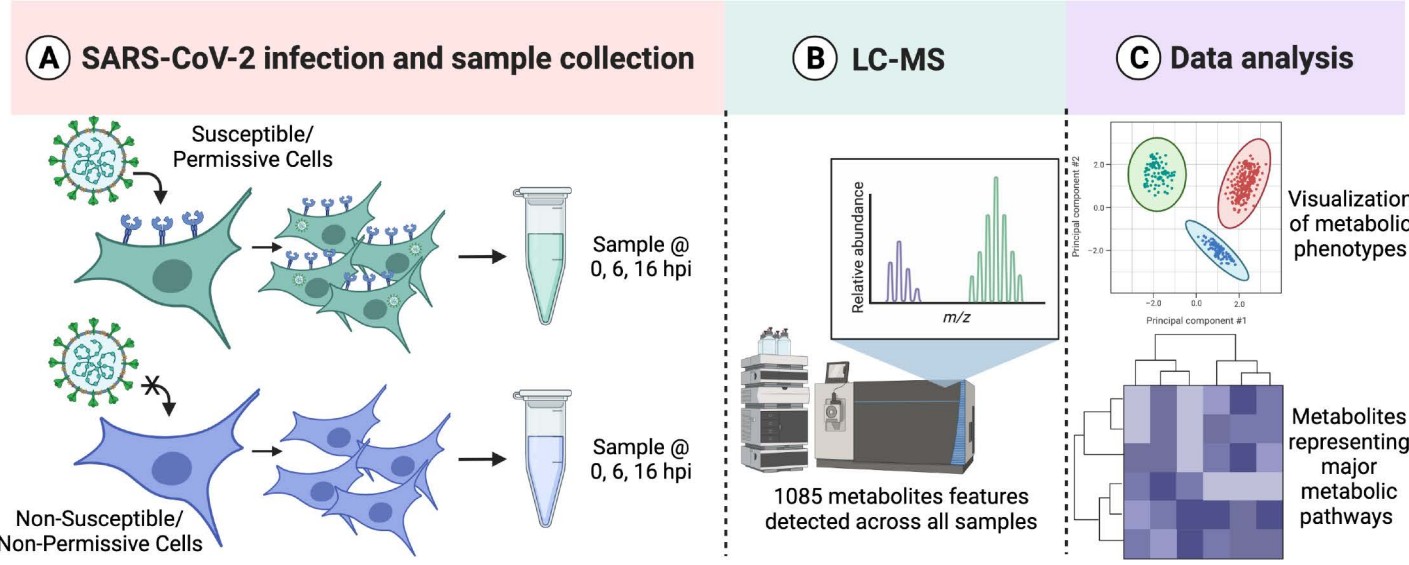

**Fig 2. Metabolic profiling analysis pipeline for infected cells.** (**A**) ACE2-A549 and A549 cells were infected with SARS-CoV-2 Isolate USA-WA1/2020 at an MOI of 10. Metabolites were extracted at multiple time points post infection. (**B**) Metabolic profiles are generated following LC-MS detection in the Mass Spectrometry Core Facility. (**C**) Data processing and analysis allows for both global metabolite profiling and pathway enrichment analysis. Image generated with Biorender.

processed for LC-MS metabolite detection (Fig 2B). Samples were analyzed via LC-MS to identify molecules smaller than ~ 1000 Da, which can include hormones, oligonucleotides, peptides, and other molecular products of cellular biochemical reactions [31,32]. A total of 1085 metabolites were detected across all samples and were included in all analyses. Data analyses were performed using MetaboAnalyst allowing for quantification of untargeted metabolites and identification of changes in the metabolomic phenotypes at each of the different time points (Fig 2C) [31–33].

## Metabolic profiling of ACE2-A549 cells during SARS-CoV-2 infection

We began by analyzing metabolic changes in ACE2-A549 cells that support productive SARS-CoV-2 infection. Changes in the global metabolomic profiles of infected ACE2-A549 cells were determined using unsupervised PCA and supervised PLS-DA (Fig 3A and B). From these analyses, the variance between each time point was greater than the variance between replicates within each group. We observed that the first 2 PLS-DA components represented 42% of the overall variance, further demonstrating that the three time points are distinct from each other. This is much greater than the expected 0.03% variance that would be expected from a uniformly random distribution of metabolites. Taken together, our data suggests that greater metabolomic changes occur over time although some overlap between samples is observed. Both analyses confirm that clear, non-random, differences exist between productively infected ACE2-A549 cells harvested at different each time points.

To further examine metabolomic patterns that significantly change during SARS-CoV-2 infection in ACE2-A549 cells, we performed ANOVA to assess changes in metabolomic between cells harvested at 0, 6, and 16 hpi. From this analysis, 152 metabolite features with an FDR-corrected p-value < 0.05 were differentially regulated between timepoint groups. Heatmap analysis of these ANOVA metabolite features revealed temporal changes in metabolite phenotypes from 0 to 16 hpi (Fig 3C). The variance between samples can also be observed when each metabolite feature is plotted for the individual samples (S2 Fig). Clustering analysis of similarly altered metabolite feature produces 4 main classes: reduced at 16 hpi (class 1), increased at 16 hpi (class 2), reduced at 6 hpi (class 3), and increased at 6 hpi (class 4). The majority of altered metabolite features, belonging to class 1, had the highest abundance at 0 hpi and progressively decreased from 6 to 16 hpi, suggesting a trajectory of depletion during the course of SARS-CoV-2 replication. In contrast, class 2 metabolite features increased in abundance from 0 to 16 hpi. For classes 3 and 4, metabolite features with detected changes at 6 hpi often returned to baseline abundance by 16 hpi.

To derive additional biological relevance, the 152 metabolite features distinguished by ANOVA were then manually searched by m/z value in METLIN to make putative metabolite identifications [35]. Identified putative metabolites were found in the four main classes (Fig 3C). Select metabolites are presented to highlight some of the metabolic changes detected during infection (Fig 4). Each boxplot depicts the normalized fold-change of the putative metabolite description for each class with average values represented as yellow diamonds and individual replicates within each time point represented as black spots. These metabolite features may be associated with some flux or alteration in utilization of intermediates in metabolic pathways.

## Distinct metabolic phenotypes of A549 cells during SARS-CoV-2 infection

We next sought to separate metabolic changes identified during productive infection from changes that may result from responses due to virus exposure. To accomplish this, we analyzed A549 cells, which do not express ACE2 and are refractory to infection at 0, 6 and 16 hpi after exposure to SARS-CoV-2 inoculum [19,20]. As before, we evaluated metabolomic

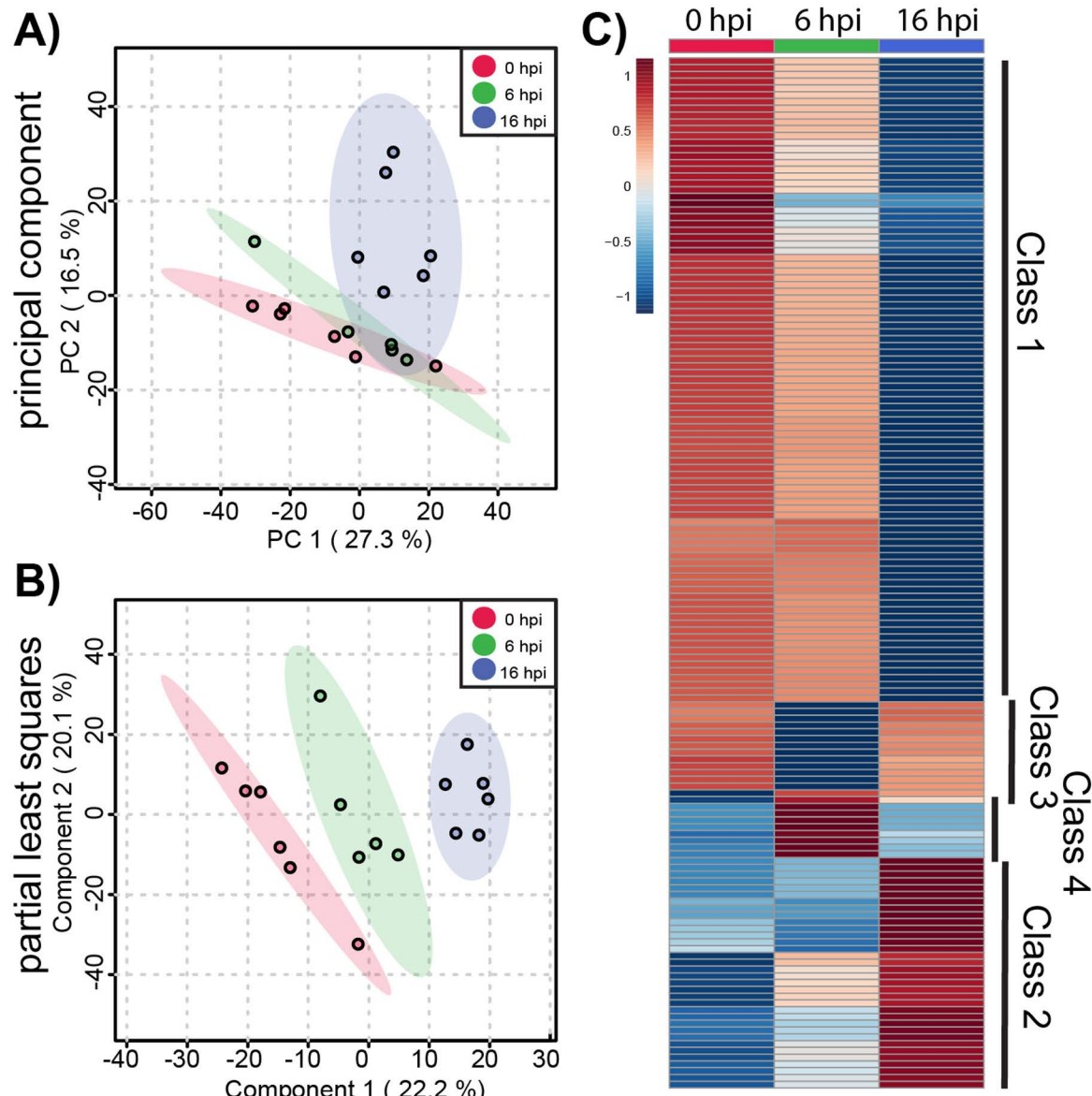

**Fig 3. Global metabolite profiling of SARS-CoV-2 infected ACE2-A549 cells.** A total of 1085 metabolites were analyzed by **(A)** principal component analysis (PCA) and **(B)** supervised partial least-squares discriminant analysis (PLSDA). **(C)** Heatmap analysis of significant metabolites (n = 152) reveals temporal changes in metabolite phenotypes from 0 to 16 hpi. Mean intensities of each metabolite were clustered into 3 groups: 0 hpi (n = 6), 6 hpi (n = 5), and 16 hpi (n = 6). Normalized fold change of specific metabolites is relative to peak concentration across 0, 6, and 16 hpi.

phenotypes using PCA and PLS-DA and observed larger ellipses that somewhat overlap. PLS-DA components 1 and 2 represented 43.2% of the overall variance in the dataset. Similar to the ACE2-A549 cells, these findings suggest that metabolomic differences between A549 cells harvested at different timepoints exist, despite no detectable infection or viral replication. (Fig 5A and B). From the inoculated A549 cells, we identified 377 metabolite features that had an ANOVA FDR-corrected p-value < 0.05. To further visualize metabolic dysregulation across 0, 6, and 16 hpi, heatmap analysis was performed (Fig 5C). The distribution and consistency of detected metabolite features across the replicate samples can be seen in the

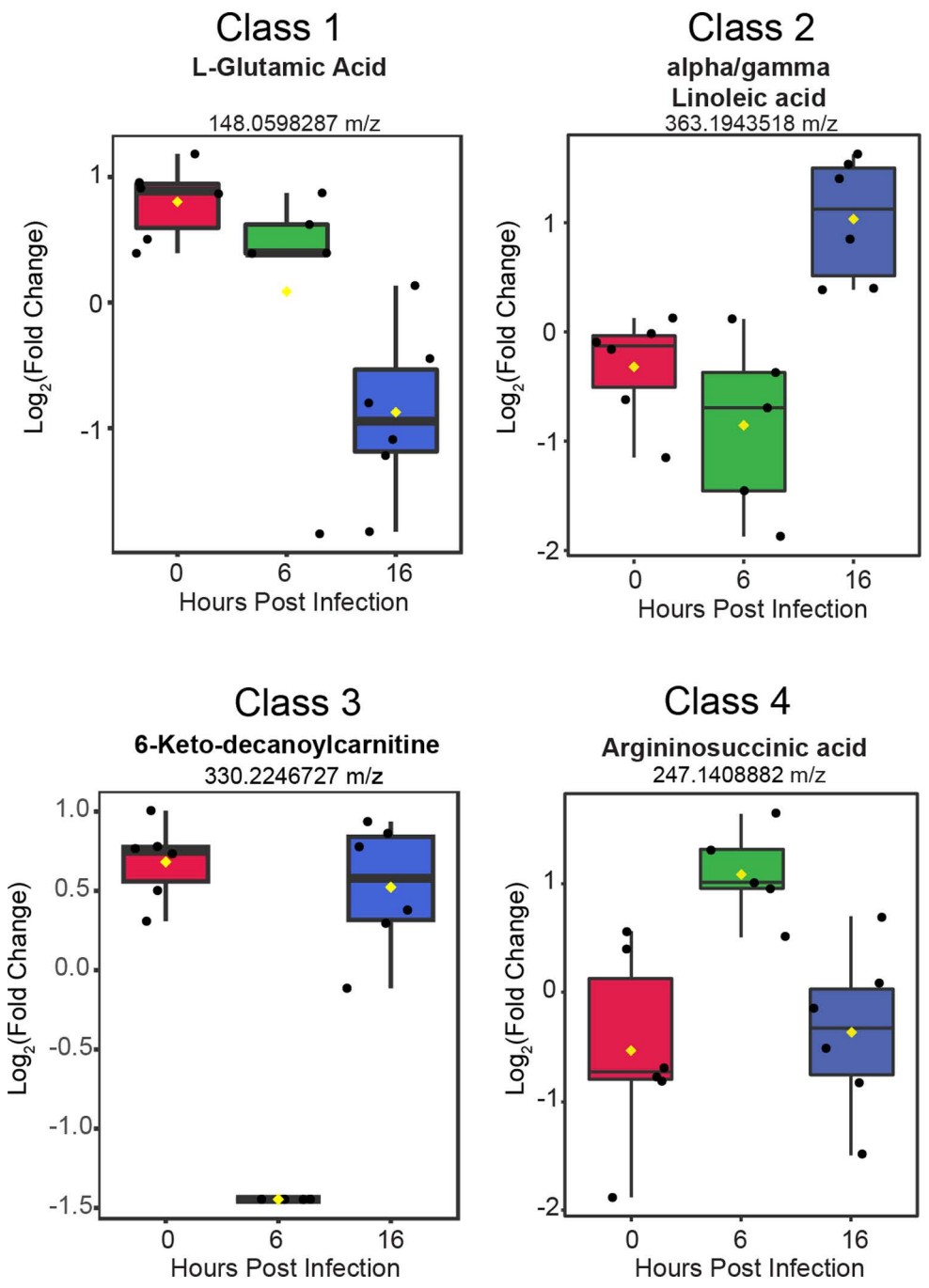

**Fig 4. Putative metabolites identified from each of the four classes.** The normalized fold change value for 4 separate metabolites are presented within box plots. Individual replicates within each time point are represented as black spots. Average value is the yellow diamond. Each plot is labeled for the individual metabolite(s) based on m/z value identification.

expanded heatmap (S3 Fig). Unlike with ACE2-A549 cells, the majority of metabolic changes are increasing quantities of metabolite features that peak at 6 hpi and remain elevated through 16 hpi (Fig 5C). The second prominent class of metabolite features exhibits a transient increase in detection at 6 hpi, followed by reduced detection at or near levels seen at 0 hpi. The

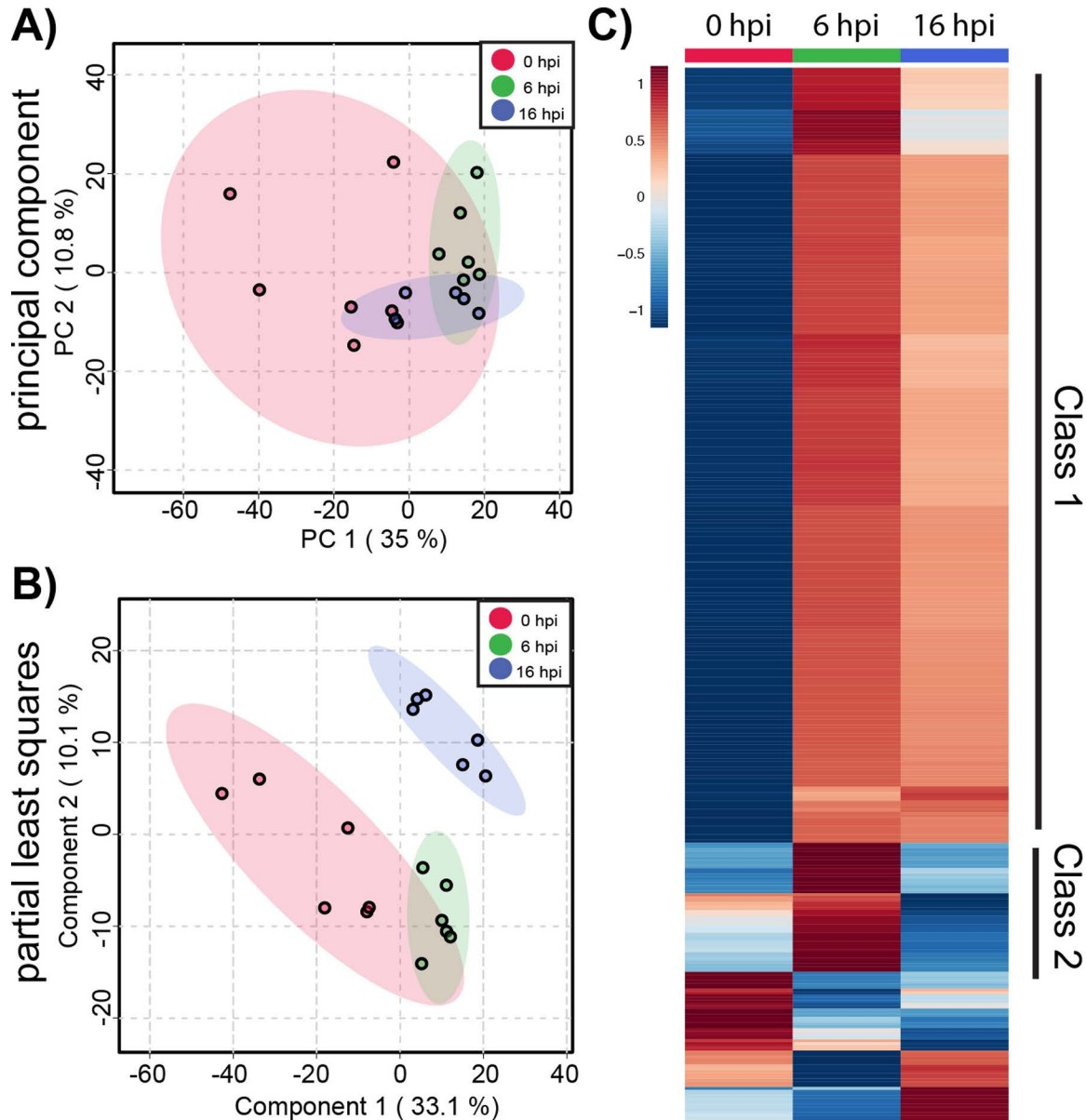

**Fig 5. Global metabolite profiling of SARS-CoV-2 inoculated A549 cells.** Similar to ACE2-A549 cells, metabolite profiles from A549 cells were compared by (**A**) PCA and (**B**) PLSDA. (**C**) Heatmap analysis of significant metabolites (n = 377) reveals a different temporal phenotype in cells that are exposed to SARS-CoV-2 but remain uninfected. Mean intensities of each metabolite were clustered into 3 groups: 0 hpi (n = 6), 6 hpi (n = 6), and 16 hpi (n = 6). Normalized fold change of specific metabolites is relative to peak concentration across 0, 6, and 16 hpi.

remaining significantly changed metabolite features vary with peak detection seen either at 0 or 16 hpi. Overall, the majority of changes in these cells likely represent changes in response to the inoculum that decrease by the later time points after inoculation.

## Comparison of metabolites between ACE2 and A549 cells

We sought to further understand the differing metabolic responses between A549 cells exposed to SARS-CoV-2 and ACE2-A549 cells infected with SARS-CoV-2. Statistically

significant features distinguished by ANOVA analyses for both comparisons were investigated to identify metabolite features that were either shared or unique between the two cell types. Of the 152 significant metabolite features from ACE2-A549 cells and 377 significant metabolite features from A549 cells, only 47 were significantly changed following SARS-CoV-2 exposure in both cell types (Fig 6A). In addition, pathway analysis of metabolite features that are dysregulated during SARS-CoV-2 replication was performed (Fig 6B).

In total, 13 metabolic pathways were altered in ACE2-A549 cells with productive SARS-CoV-2 replication. A majority, 10 pathways, were involved in amino acid metabolism including alanine, aspartate, cysteine, glutamate, glycine, histidine, lysine, methionine, and threonine (Table 1).

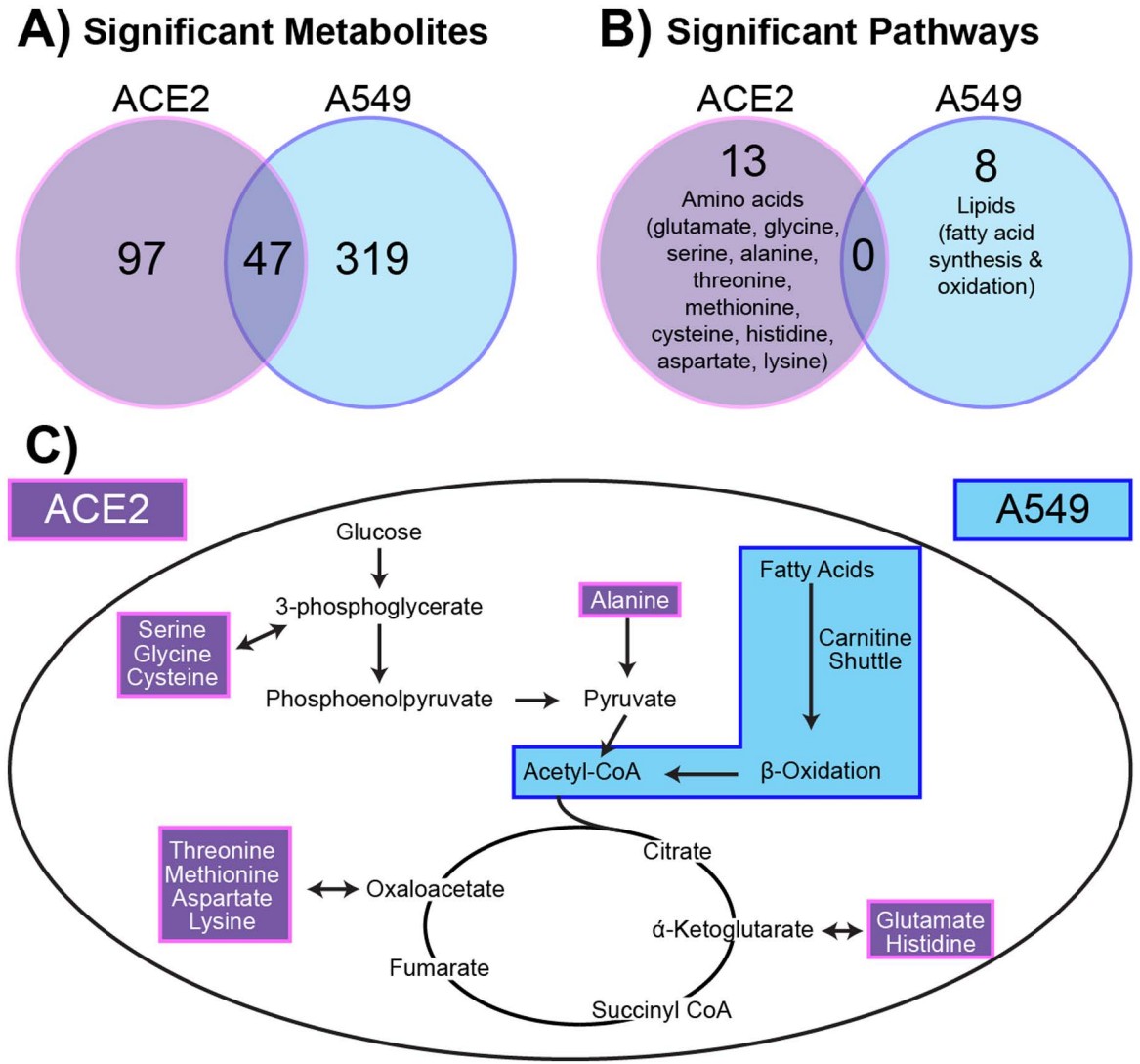

**Fig 6. A549 cells have quantifiably different metabolic responses to SARS-CoV-2. (A)** Significant metabolites (FC > 1.5) from ACE2-A549 and A549 cells were compared to identify similar and different metabolites that change in response to SARS-CoV-2. **(B)** Pathway analysis on significant metabolites from ACE2-A549 and A549 cells. **(C)** Evaluating metabolic pathway changes relative to energy generation. Amino acids that were downregulated in ACE2-A549 are shown within the pathway highlighted in purple. Fatty acid metabolism was upregulated in A549 cells exposed to SARS-CoV-2, highlighted separately in blue. Both pathways directly or indirectly connect with components of the TCA cycle and glycolysis.

**Table 1. Altered metabolic pathways during SARS-CoV-2 infection of ACE2-A549 cells.**

| ACE2-A549 Pathways | Pathway total | Hits Total | Hits Sig | Gamma Value | Cpd.Hits |
|---|---|---|---|---|---|
| Glutamate metabolism | 15 | 5 | 5 | 0.00558 | L-Glutamic acid \| Oxoglutaric acid \| Gamma-Aminobutyric acid \| Glutathione \| Succinic acid semialdehyde |
| Glutathione Metabolism | 19 | 4 | 4 | 0.009278 | Pyroglutamic acid \| L-Glutamic acid \| Glutathione \| L-Cysteine |
| Tryptophan metabolism | 94 | 9 | 7 | 0.009861 | L-Glutamic acid \| Oxoglutaric acid \| Formyl-5-hydroxykynurenamine \| 2-Aminobenzoic acid \| CE2095 \| L-Kynurenine \| Glutathione |
| Glycine, serine, alanine and threonine metabolism | 88 | 8 | 6 | 0.015295 | L-Glutamic acid \| Oxoglutaric acid \| Dimethylglycine \| Creatine \| 2-Ketobutyric acid \| L-Allothreonine \| L-Threonine \| Glutathione |
| Methionine and cysteine metabolism | 94 | 8 | 6 | 0.015295 | L-Glutamic acid \| Oxoglutaric acid \| 2-Ketobutyric acid \| DL-Glutamate \| L-Cysteine \| Allocystathionine \| L-Cystathionine \| Glutathione |
| Urea cycle/amino group metabolism | 85 | 10 | 7 | 0.018261 | L-Glutamic acid \| Creatinine \| Creatine \| Argininosuccinic acid \| Gamma-Aminobutyric acid \| Oxoglutaric acid \| Queuine \| N4-Acetylaminobutanal |
| Histidine metabolism | 33 | 3 | 3 | 0.01834 | L-Glutamic acid \| Oxoglutaric acid \| Glutathione |
| Beta-Alanine metabolism | 20 | 3 | 3 | 0.01834 | L-Glutamic acid \| Gamma-Aminobutyric acid \| Oxoglutaric acid |
| Alanine and Aspartate Metabolism | 30 | 4 | 3 | 0.041655 | L-Glutamic acid \| Oxoglutaric acid \| N-Acetyl-L-aspartic acid \| Argininosuccinic acid |
| Lysine metabolism | 52 | 4 | 3 | 0.041655 | L-Glutamic acid \| Oxoglutaric acid \| Pipecolic acid |
| Glycerophospholipid metabolism | 156 | 6 | 4 | 0.042495 | Alpha-Linolenic acid \| Glycerophosphocholine \| Phosphorylcholine \| Linoleic acid |
| C5-Branched dibasic acid metabolism | 10 | 2 | 2 | 0.045209 | Itaconic acid \| Mesaconic acid |
| Ascorbate (Vitamin C) and Aldarate Metabolism | 29 | 2 | 2 | 0.045209 | Glutathione \| L-Erythrulose |

Additional metabolic pathways detected included glycerophospholipid metabolism, C5-branched dibasic acid metabolism, and ascorbate metabolism. Identifying if these altered pathways are caused by productive viral replication cannot be assessed from this data alone. We compared pathway changes in cells that cannot be productively infected by SARS-CoV-2. Performing pathway analysis on the 377 metabolite features distinguished by ANOVA from the A549 cells inoculated with SARS-CoV-2 identified 8 distinct pathways. Curiously, none of these pathways overlap with those identified in from ACE2-A549 analysis (Fig 6B). The 8 pathways unique to A549 cells mapped exclusively to lipid metabolism and included: fatty acid oxidation, activation, and metabolism; di-unsaturated fatty acid beta-oxidation, de novo fatty acid biosynthesis, omega-3 fatty acid metabolism, and carnitine shuttle (Table 2). Interestingly, metabolite features associated with immunomodulatory leukotrienes were also detected. Overall, these extensive changes to lipids, specifically fatty acids, suggest a change not only towards an inflammatory state, but also a shift in energy source by the cells following exposure to a non-productive infection.

Taken together, changes in metabolism between two cell lines over a 16-hour period of exposure to SARS-CoV-2 demonstrate metabolomic differences in a range of individual metabolites and pathways. Specifically, amino acid related-pathways were dysregulated in ACE2-A540 while lipid-related pathways were dysregulated in A549 cells exposed to SARS-COV2 (Fig 6C).

## Discussion

In this study, we sought to understand the nature of cellular metabolic shifts in response to SARS-CoV-2 infection. To distinguish changes associated with viral replication from exposure to infectious virus, we compared ACE2-expressing A549 cells that are susceptible and support productive viral replication with A549 cells that are not susceptible to infection. We chose time points that represent early, intermediate, and late stages of viral replication to

**Table 2. Altered metabolic pathways following exposure of A549 cells to SARS-CoV-2.**

| A549 Pathways | Pathway total | Hits Total | Hits Sig | Gamma Value | Cpd.Hits |
|---|---|---|---|---|---|
| Di-unsaturated fatty acid beta-oxidation | 26 | 5 | 5 | 0.00594 | CE2422 \| CE2421 \| CE2434 \| CE0849 \| Linoleic acid |
| Fatty acid activation | 74 | 9 | 6 | 0.011166 | Palmitoyl-CoA \| Alpha-Linolenoyl-CoA \| Alpha-Linolenic acid \| Gamma-Linolenic acid \| Linoleic acid \| Heptadecanoyl-CoA |
| Fatty Acid Metabolism | 63 | 4 | 3 | 0.024423 | 3-Oxohexadecanoyl-CoA \| Palmitoyl-CoA \| Linoleic acid |
| Fatty acid oxidation | 35 | 4 | 3 | 0.024423 | Palmitoyl-CoA \| Alpha-Linolenoyl-CoA \| Heptadecanoyl-CoA |
| De novo fatty acid biosynthesis | 106 | 7 | 4 | 0.031172 | Palmitoyl-CoA \| Alpha-Linolenic acid \| Gamma-Linolenic acid \| Linoleic acid |
| Leukotriene metabolism | 92 | 7 | 4 | 0.031172 | 20-Hydroxy-leukotriene B4 \| CE6473 \| CE6228 \| CE6182 \| CE6187 |
| Omega-3 fatty acid metabolism | 39 | 2 | 2 | 0.034719 | trans-2-Enoyl-OPC8-CoA \| Alpha-Linolenic acid |
| Carnitine shuttle | 72 | 5 | 3 | 0.040388 | Alpha-Linolenoyl-CoA \| Palmitoyl-CoA \| Heptadecanoyl-CoA |

evaluate the temporal changes in metabolites following inoculation. Our metabolic pathway analysis found 152 and 377 significantly changed metabolites in ACE2-A549 and A549 cells, respectively. Surprisingly, there was limited overlap in altered metabolites or altered metabolic pathways between cells undergoing productive infection and those exposed to infectious virus. Critically, we identified alterations in pathways that are potentially involved in either productive viral infection or in cellular anti-viral responses to SARS-CoV-2.

## Consequences of productive viral infection

The initial focus of our analysis was the changes to cellular metabolism induced by active viral replication. The ACE2-A549 cells are a widely used model that we observed to be both susceptible to SARS-CoV-2 infection and permissive for productive viral replication [19,20]. We analyzed metabolic shifts immediately after virion entry (0 hpi), a mid-point of viral replication prior to virion production (6 hpi), and a late timepoint when new virions are being released from infected cells (16 hpi) [34]. Overall, we identified four different classes of metabolites based on the relative increase or decrease in detection between each time point. We used these changes to identify pathways that were altered by active SARS-CoV-2 replication. Most notably, we observed that the majority of altered metabolic pathways were associated with amino acid metabolism.

Within the identified pathways, we identified L-Glutamic Acid as a major putative metabolite that was significantly reduced from 0 to 16 hpi. A previous study demonstrated that SARS-CoV-2 infection rewires carbon entry into the TCA cycle [17]. Mullen et al. showed that oxidative metabolism of glutamine through the TCA cycle was reduced during SARS-CoV-2 infection in favor of pyruvate utilization via pyruvate carboxylase [17]. This shift increased levels of oxaloacetate and also maintain synthesis of aspartate, which is used to synthesize pyrimidine nucleotides [36,37]. Interestingly, both glutamate and aspartate metabolic pathways were considered significant in our analysis of SARS-CoV-2 infected ACE2-A549 cells.

In addition to changes to L-glutamic acid and aspartate, the metabolic shifts observed in SARS-CoV-2 infected ACE2-A549 cells reflect similar changes observed in COVID-19 positive patient serum samples [10,23,27,38–43]. It is notable that our cell culture model identified similar metabolic pathways being disrupted during SARS-CoV-2 infection, even at 6 hpi. This indicates that metabolic screening of different laboratory-based model systems may be able to accurately generate data on potential biomarkers for SARS-CoV-2 and potentially other infectious diseases.

## Metabolites responding to viral inoculation

While metabolic changes due to active viral replication are important, not all cells within a tissue or organ system are equally susceptible to virus infection. Thus, we hypothesized that uninfected cells that do not support SARS-CoV-2 infection can respond to virus exposure with altered metabolism leading to further metabolic dysfunction. To test this hypothesis, we analyzed the metabolic profile of A549 cells that do not support viral entry following exposure to the same infectious virus inoculum as the ACE2-A549 cells. Our data confirmed that lack of ACE2 expression in A549 cells resulted in a complete lack of infection and replication following SARS-CoV-2 inoculation. While we have no evidence of viral replication in A549 cells, the resulting changes in the metabolic profile of these cells indicates that they are responding to the virus inoculum. Specifically, fatty acid catabolic (β-oxidation) and anabolic (de novo fatty acid synthesis) pathways were significantly altered following A549 cell exposure to infectious virus. Lipid dysregulation has been a hallmark of COVID-19 pathology in patients and a hallmark of disease severity and progression [41,44,45]. Both pathways converge on acetyl-CoA, a critical molecule in the breakdown of fatty acids and the synthesis of other lipid types, such as cholesterol, which can be transformed into other steroids with pro and anti-inflammatory mechanisms [46,47]. This is further supported by the significant number of identified metabolites associated with leukotriene and omega-3 fatty acid metabolism within the A549 cells. Increased leukotriene production is connected to COVID-19 through transcriptional and metabolic studies from patient serum and infected monocytes [48–50]. Another significant metabolite in our profile of the A549 cells was palmitoyl-CoA, a major component in the synthesis of ceramide and sphingolipids [51]. Previous studies of COVID-19 patient serum samples found distinct increases in sphingosine and ceramides [52,53]. These increases were distinct between patients with mild disease and those in intensive care [53]. The data suggests that uninfected cells respond to SARS-CoV-2, altering metabolic profiles and possibly increasing the production of pro- or anti-inflammatory biomolecules and enzyme cofactors [3,4,54]. Thus, even uninfected cells may be contributing to the overall pathology observed in COVID-19 patients.

## Conclusion

Cellular models for SARS-CoV-2 infection are incredibly important for the initial testing of interventions that directly target viral replication. Through our metabolomic profiling, we identified metabolites and metabolic profiles that are associated with both active viral infection and exposure to infectious virus. Our analysis identified a range of metabolites and metabolic pathways altered by productive viral replication. Further work will be required to understand if these metabolites promoting viral replication are specific for SARS-CoV-2 infection and the relationship to changes in primary cells and organs. While cellular metabolism is often thought to be manipulated by the virus for its own ends, it is also connected to antiviral responses [1–3]. Cells can produce antiviral metabolites or inhibit metabolic pathways to hinder viral replication [3]. As with productive replication, further experiments that either promote shifts in leukotrienes or other inflammatory molecules will need to be performed to characterize their effects on SARS-CoV-2 infection. It is also possible that the identified metabolite profiles can be developed as biomarkers of infection that could be used for surveillance testing or as a predictive tool for risk evaluation of severe disease. The similarities of our results to metabolic shifts observed in patients suggest a potential platform for methodological development. Discriminatory metabolites defining infection can be correlated to patient metabolic profiles to facilitate our understanding of SARS-CoV-2 induced pathologies. Through both the immediate findings and the development of more complex models, we

hope to increase our understanding of how SARS-CoV-2 replication and spread correlates to disease. Through that understanding, we can then find better therapeutics to limit morbidity and mortality from COVID-19.

## Supporting Information

**S1 Fig. Cellular morphology during productive and non-productive SARS-CoV-2 infection.** A) Further images of ACE2-A549 and A549 cells from Fig 1B. Presented at three-channel merged images of the transmitted light (greyscale), SARS-CoV-2 anti-nucleocapsid antibody (red), and Dapi (blue, nuclei). Scale bar 100 μm. B) Further images of infected ACE2-A549 and A549 cells at 16hpi as in Fig 1C. The three channel merged images are of SARS-CoV-2 anti-nucleocapsid antibody (red), actin filaments stained with phalloidin (green) and Dapi (blue, nuclei). Scale bars are 10 μm.
(EPS)

**S2 Fig. Heatmap analysis of significant metabolites (n = 152) for all ACE2-A549 samples at each time point.**
(EPS)

**S3 Fig. Heatmap analysis of significant metabolites (n = 377) for all A549 samples at each time point.**
(EPS)

**S1 Table. Raw mass spec values for all samples and timepoints.**
(XLSX)

## Acknowledgments

The following reagent was obtained through BEI Resources, NIAID, NIH: Human Lung Carcinoma Cells (A549) Expressing Human Angiotensin-Converting Enzyme 2, NR-53821. The following reagent was deposited by the Centers for Disease Control and Prevention and obtained through BEI Resources, NIAID, NIH: SARS-Related Coronavirus 2, Isolate USA-WA1/2020, NR-52281. The content is solely the responsibility of the authors and does not necessarily represent the official views of the National Institutes of Health.

## Author contributions

**Conceptualization:** Emma K. Loveday, Ronald K. June, Matthew P. Taylor.

**Data curation:** Emma K. Loveday, Hope Welhaven, Ayten Ebru Erdogan, Kyle S. Hain, Matthew P. Taylor.

**Formal analysis:** Emma K. Loveday, Hope Welhaven, Ayten Ebru Erdogan, Luke F. Domanico, Matthew P. Taylor.

**Funding acquisition:** Connie B. Chang, Ronald K. June, Matthew P. Taylor.

**Investigation:** Emma K. Loveday, Hope Welhaven, Kyle S. Hain, Luke F. Domanico, Matthew P. Taylor.

**Methodology:** Emma K. Loveday, Hope Welhaven, Ayten Ebru Erdogan, Luke F. Domanico, Ronald K. June, Matthew P. Taylor.

**Project administration:** Emma K. Loveday, Matthew P. Taylor.

**Resources:** Matthew P. Taylor.

**Software:** Ayten Ebru Erdogan.

**Supervision:** Emma K. Loveday, Connie B. Chang, Ronald K. June, Matthew P. Taylor.

**Validation:** Emma K. Loveday, Hope Welhaven, Matthew P. Taylor.

**Visualization:** Emma K. Loveday, Matthew P. Taylor.

**Writing – original draft:** Emma K. Loveday, Hope Welhaven, Ayten Ebru Erdogan, Kyle S. Hain, Ronald K. June, Matthew P. Taylor.

**Writing – review & editing:** Emma K. Loveday, Hope Welhaven, Ayten Ebru Erdogan, Connie B. Chang, Ronald K. June, Matthew P. Taylor.

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
