## [Decision Letter · Decision Letter 0]

17 Jun 2024

PONE-D-24-11682Starve a cold or feed a fever?

Identifying cellular metabolic changes following infection and exposure to SARS-CoV-2PLOS ONE

Dear Dr. Loveday,

Thank you for submitting your manuscript to PLOS ONE. After careful consideration, we feel that it has merit but does not fully meet PLOS ONE’s publication criteria as it currently stands. Therefore, we invite you to submit a revised version of the manuscript that addresses the points raised during the review process.

We look forward to receiving your revised manuscript.

Kind regards,

Arunava Roy, Ph.D.

Academic Editor

PLOS ONE

Journal Requirements:

2. Thank you for stating the following financial disclosure: "RKJ

NIH/NIAMS R01AR073964 and R01AR081489

National Institutes of Health, National Institute of Arthritis, Musculoskeletal, and Skin Diseases

https://www.niams.nih.gov/

No role

RKJ

NSF CMMI 1554708

National Science Foundation, Division of Civil, Mechanical and Manufacturing Innovation

https://www.nsf.gov/div/index.jsp?div=CMMI

no role

CBC

NIH/NIAID 1R56AI156137-01

National Institutes of Health, National Institute of Allergy and Infectious Disease

https://www.niaid.nih.gov/

No role

MPT

Montana State University VPREDGE COVID Research Award   

Internal Award

https://www.montana.edu/research/

No role      " 

3. Thank you for stating the following in the Acknowledgments Section of your manuscript: "The authors acknowledge funding from NSF (CMMI 1554708), NIH (NIAMSR01AR073964 and R01AR081489 and NIAID 1R56AI156137-01), and a Montana State University VPREDGE COVID Research Award. The following reagent was obtained through BEI Resources, NIAID, NIH: Human Lung Carcinoma Cells (A549) Expressing Human Angiotensin-Converting Enzyme 2, NR-53821. The following reagent was deposited by the Centers for Disease Control and Prevention and obtained through BEI Resources, NIAID, NIH: SARS-Related Coronavirus 2, Isolate USA-WA1/2020, NR416 52281. Funding for the Montana State Mass Spectrometry Facility used in this publication was made possible in part by the MJ Murdock Charitable Trust, the National Institute of General Medical Sciences of the National Institutes of Health under Award Numbers P20GM103474 and S10OD28650, and the MSU Office of Research and Economic Development. The content is solely the responsibility of the authors and does not necessarily represent the official views of the National Institutes of Health."

Please remove any funding-related text from the manuscript and let us know how you would like to update your Funding Statement. Currently, your Funding Statement reads as follows: "RKJ

NIH/NIAMS R01AR073964 and R01AR081489

National Institutes of Health, National Institute of Arthritis, Musculoskeletal, and Skin Diseases

https://www.niams.nih.gov/

No role

RKJ

NSF CMMI 1554708

National Science Foundation, Division of Civil, Mechanical and Manufacturing Innovation

https://www.nsf.gov/div/index.jsp?div=CMMI

no role

CBC

NIH/NIAID 1R56AI156137-01

National Institutes of Health, National Institute of Allergy and Infectious Disease

https://www.niaid.nih.gov/

No role

MPT

Montana State University VPREDGE COVID Research Award   

Internal Award

https://www.montana.edu/research/

No role      "

Reviewers' comments:

Reviewer's Responses to Questions

**Comments to the Author**

1. Is the manuscript technically sound, and do the data support the conclusions?

Reviewer #1: Yes

Reviewer #2: Yes

2. Has the statistical analysis been performed appropriately and rigorously? 

Reviewer #1: Yes

Reviewer #2: Yes

3. Have the authors made all data underlying the findings in their manuscript fully available?

Reviewer #1: Yes

Reviewer #2: Yes

4. Is the manuscript presented in an intelligible fashion and written in standard English?

Reviewer #1: Yes

Reviewer #2: Yes

5. Review Comments to the Author

Reviewer #1: In this study, Loveday et al characterize the metabolic profile of a ACE2-expressing modified human lung cell line on exposure to SARS-CoV-2 virus. An unbiased metabolic profiling will be of interest to the broad audience of SARS-CoV-2 biology and PLOS one audience. Overall, the results presented and the conclusions derived are supported by the data collected by the authors. The authors also give context to the study in the results and conclusion section. However, before accepting this manuscript there are a few additions that need to be made that will greatly improve the rigor of the science put forward by the authors. I am noting these comments/additions below:

1. Since most of the results are drawn based on the Ace2 expressing cell lines, it will be good to confirm Ace2 expression in the cell lines used

2. Authors proactively discuss the health of cells on virus exposure and check for actin filaments. It would however be good to elaborate on this, either by providing more images in the supplementary: more replicates, image for 6h exposure, show what different morphologies of cells were observed (currently only 1 image is given) or testing overall cell viability. This will help the readers understand if MOI of 10 is cytotoxic in any way and draw conclusions with appropriate context

3. It is unclear how many replicates were used for each experiment. Were they biological replicates or technical, i.e., grown in the same well and collected at different time points or virus exposure was done in separate wells. Elaborating this in the methods and in figure legend would be useful

4. More details on how the replicates were used for the final pathway analysis is needed, were the data aggregated or something else?

5. Supplementary data file (Excel) needs description for the various columns and abbreviations used and at what step of analysis were they generated

Reviewer #2: The authors of this paper try to figure out the metabolic change of SARS-CoV infection using A549 cells. Overall it is a well-designed study with interesting results. If possible, the following concerns about the paper should be adjusted to further prove the metabolic change they observed with SARS-CoV infection.

1. Why did the team choose such high MOI to setup the experiment? My concern is anything observed here might be just a high dose artificial effect. What is the MOI level/titer levels that usually happens in SARS-CoV patient infection?

2. Figure 1C the extracellular viral titer of A549 ACE2 cell lines in 6hpi does not show any error bar/data point, so not clear to me if it is really lower than 0h and if it is lower than the A549 control cells

3. Is it possible to verify the most altered amino acid pathways in human primary lung cells SARS-CoV infection setting?

6. PLOS authors have the option to publish the peer review history of their article (what does this mean? ). If published, this will include your full peer review and any attached files.

**Do you want your identity to be public for this peer review?** For information about this choice, including consent withdrawal, please see our Privacy Policy .

Reviewer #1: No

Reviewer #2: No

---

## [Author Response · Author response to Decision Letter 1]

26 Nov 2024

Thank you for the review of the manuscript and the opportunity to revise for resubmission. The reviewers were very positive about our study, complimenting both study-design and relevant, interesting findings. We greatly appreciated the constructive feedback. To address the specific comments, we have included Western detection of ACE2 expression, additional microscopy images of the infected cells, and provided further clarification of our experimental design and interpretation. We believe the changes have addressed the concerns and strengthened the overall manuscript.

Reviewer #1: In this study, Loveday et al characterize the metabolic profile of a ACE2-expressing modified human lung cell line on exposure to SARS-CoV-2 virus. An unbiased metabolic profiling will be of interest to the broad audience of SARS-CoV-2 biology and PLOS one audience. Overall, the results presented and the conclusions derived are supported by the data collected by the authors. The authors also give context to the study in the results and conclusion section. However, before accepting this manuscript there are a few additions that need to be made that will greatly improve the rigor of the science put forward by the authors. I am noting these comments/additions below:

1. Since most of the results are drawn based on the Ace2 expressing cell lines, it will be good to confirm Ace2 expression in the cell lines used

We appreciate the positive consideration of the work and for your concern. We agree that the expression of ACE2 is a fundamental property that defines infection and is the difference between the two cells lines. To confirm the cells that were acquired for our studies did express different quantities of ACE2, we performed a western blot against ACE2, with an actin loading control, from cell extracts of both lines. This data has been appended to figure 1 as part A to complement the characterization of both cell types. Additional methods to support this work are appended in the materials and methods section (line 115-127). The results are also briefly addressed in the results (line 195-197).

2. Authors proactively discuss the health of cells on virus exposure and check for actin filaments. It would however be good to elaborate on this, either by providing more images in the supplementary: more replicates, image for 6h exposure, show what different morphologies of cells were observed (currently only 1 image is given) or testing overall cell viability. This will help the readers understand if MOI of 10 is cytotoxic in any way and draw conclusions with appropriate context.

Thank you for appreciating the value of the experiments and requesting further information regarding cell health. To best address the reviewer concern, we have built a supplemental figure with lower magnification images of the infected cells at T6 and T16, stained for viral antigen that includes a phase contrast image within the fluorescent channel merge. The images depict cellular morphologies and nuclear distributions of “intact” cells. The depicted shapes and density are consistent with what was observed at the time of infection. We have also included a representative T16 A549 image stained with phalloidin and another example of T16 ACE2 (line 214-217).

We are unsure what the reviewer means as “cytotoxic” with regards to the cell viability. The productively infected ACE2-cells will undergo rounding and detachment as infection progresses, whereas the non-susceptible A549 cells do not. Our experimental time points were meant to sample cells prior to the onset of cellular rounding indicative of terminal time points. As we note in the text, other experiments and published literature using longer time courses do show cell rounding consistent with end-stage viral replication.

3. It is unclear how many replicates were used for each experiment. Were they biological replicates or technical, i.e., grown in the same well and collected at different time points or virus exposure was done in separate wells. Elaborating this in the methods and in figure legend would be useful.

Thank you for identifying the need for clarification of this point. This is a critical element to appreciate the results of our work. We have modified the materials and methods (lines 149-151 and 178-188) (see text below) to highlight the sample replication. Cells of each type were seeded in 6 replicate wells and then infected and harvested separately, at each time point. We consider the parallel infections that were collected, extracted, and analyzed separately as biological replicates. Each sample was then measured separately. In this way, we would capture the intersample variation that is greater than the intrasample, technical, measurement error.

4. More details on how the replicates were used for the final pathway analysis is needed, were the data aggregated or something else?

We have added the text below to the materials and methods section (lines 178-188) manuscript to clarify sample sizes and pathway analysis.

For metabolomic data and downstream pathway analyses, there were 35 samples total (6 samples per timepoint for each cell line except for only 5 samples for the t6 timepoint in ACE2 cells). In total, 1,085 metabolite features were co-detected across all samples. To examine differences in regulation patterns across timepoints, ANOVA analysis was performed. The results of this analysis are that 152 and 372 metabolite features were differentially regulated across timepoints in ACE2-A549 cells and A549 cells alone, respectively. From here, we took these differentially-regulated features and performed pathway enrichment analyses. Thus 152 features were used for ACE2-A549-pathways and 372 features for A549 pathways. For features that are differentially regulated across timepoints of ACE2-A549 cells, 13 pathways were identified. Conversely, 8 pathways were differentially regulated across timepoints of A549 cells.

5. Supplementary data file (Excel) needs description for the various columns and abbreviations used and at what step of analysis were they generated

We have updated the supplementary data file to include improved descriptions for columns and abbreviations. No revision to the text is required.

Reviewer #2: The authors of this paper try to figure out the metabolic change of SARS-CoV infection using A549 cells. Overall it is a well-designed study with interesting results. If possible, the following concerns about the paper should be adjusted to further prove the metabolic change they observed with SARS-CoV infection.

1. Why did the team choose such high MOI to setup the experiment? My concern is anything observed here might be just a high dose artificial effect. What is the MOI level/titer levels that usually happens in SARS-CoV patient infection?

Thank you for the compliment on study design. We agree that the results are quite interesting and look forward to future pursuit of these results. For these experiments, MOI was a critical choice in the design and interpretation. We carefully selected this MOI not to represent initial infection in the patient but to ensure homogenous infection and exposure of all cells to sufficient infectious inoculum. The homogeneity of the response across the population of cells was critical for the subsequent extraction and detection of metabolites (see results line 202-203 and lines 236-237). Lower MOI’s of 1 and 0.1 produced heterogenous populations of infected and uninfected cells. This heterogeneity would complicate the interpretation of bulk metabolite measurements by minimizing metabolite changes between the cells that would be detected in each sample.

It is a valid concern regarding what MOI lung cells might be infected with. Initial infection of the lung could potentially be due to low numbers of virus, but likely produces high quantities of infectious virus for subsequent rounds of infection. Sampling of infectious SARS-CoV-2 in environmental and patient settings can range from as little as 5 PFU/mL up to 10^6 PFU/mL. (Lin et al., Scientific Reports (2022), 12 article 5418 https://www.nature.com/articles/s41598-022-09218-5). Our interpretation of metabolic responses would likely more resemble that seen at fulminant stages of infection of the lung when cells are exposed to higher infectious doses. Many of the published clinical analyses of patient metabolite measurements that we compare to are taken at later stages of infection, or even during recovery. We have not included this information as we feel it would be too much of an extrapolation for the discussion.

2. Figure 1C the extracellular viral titer of A549 ACE2 cell lines in 6hpi does not show any error bar/data point, so not clear to me if it is really lower than 0h and if it is lower than the A549 control cells

The measured number of plaques from the titering of those samples were at the limit of detection for the assay (1 plaque per well or 500 pfu/mL) (see materials and methods line 109-110) and no error bar can be calculated. Secondly, this is the “extracellular” sample of the productively infected cell type. Whether it is lower compared to the non-susceptible cells is not significant for our interpretation. We are merely highlighting that the T6 timepoint is before new virion production and release.

3. Is it possible to verify the most altered amino acid pathways in human primary lung cells SARS-CoV infection setting?

At this time, it is not possible for us to verify changes in primary lung cells. And this is an excellent point that we have included in the discussion (line 419) as necessary follow-up studies.

---

## [Editor Report · Decision Letter 1]

4 Dec 2024

Starve a cold or feed a fever?

Identifying cellular metabolic changes following infection and exposure to SARS-CoV-2

PONE-D-24-11682R1

Dear Dr. Loveday,

We’re pleased to inform you that your manuscript has been judged scientifically suitable for publication and will be formally accepted for publication once it meets all outstanding technical requirements.

Kind regards,

Arunava Roy, Ph.D.

Academic Editor

PLOS ONE
---

## [Editor Report · Acceptance letter]

PONE-D-24-11682R1

PLOS ONE

Dear Dr. Loveday,

I'm pleased to inform you that your manuscript has been deemed suitable for publication in PLOS ONE. Congratulations! Your manuscript is now being handed over to our production team.

Kind regards,

on behalf of

Dr. Arunava Roy

Academic Editor

PLOS ONE